# Early-onset gynecological tumors in DNA repair-deficient xeroderma pigmentosum group C patients: a case series

Andrey. A. Yurchenko[1], Brice Fresneau [2,3], Bruno Borghese[4,5,6], Fatemeh Rajabi[1], Zora Tata[7], Catherine Genestie[8], Alain Sarasin[9] & Sergey I. Nikolaev [1✉]

## Abstract

**Background** Xeroderma pigmentosum (XP) is a group of rare hereditary disorders with highly increased risk of skin tumors due to defective DNA repair. Recently we reported 34-fold increased risk of internal tumors in XP patients in comparison with general population. The molecular data and clinical practice on the internal tumors treatment in XP patients is limited and scarcely represented in the medical literature. In this work, we describe young patients with constitutive biallelic deactivation of the *XPC* gene developing gynecological tumors with somatic *DICER1* mutations.

**Methods** Whole genome sequencing was used to analyze in detail somatic mutational landscape and driver events of these rare tumors.

**Results** We describe five early-onset gynecological tumors in four xeroderma pigmentosum group C (XP-C) young patients (11 to 19 years old) including vaginal embryonal rhabdo-myosarcomas in monozygotic twin sisters, juvenile granulosa-cell tumor of the ovary and poorly differentiated stage IA Sertoli-Leydig cell tumor in 19-years old patient, and FIGO stage IC1 tumor of ovary in 13-years old patient. XP-C ovarian tumors harbor 4.4 times more single base substitutions than sporadic tissue-matched cancers and demonstrate XP-C specific mutation signature with strong transcriptional bias indicating inability of the cells to repair bulky DNA lesions of unknown etiology. A special mode of treatment was applied to avoid usage of chemotherapy which is toxic for XP patients.

**Conclusions** XP-C status should be accounted for prevention and specific treatment of gynecological tumors in young DNA repair-deficient XP patients.

## Plain language summary

Xeroderma pigmentosum group C (XP-C) is a rare inherited disorder resulting in a highly increased risk of skin and internal cancers due to the inability to efficiently repair DNA. In this study, we described four young XP-C patients who developed early-onset tumors affecting the female reproductive organs. We describe how we cared for these patients in the clinic. We looked at the genetic material within the tumors to better understand the mechanisms through which these tumors developed. We observed high numbers of specific types of changes in DNA, which are not typical for sporadic (non-inherited) gynecological tumors, but are characteristic of internal XP-C tumors. Further studies are needed to better understand the nature of these changes. Our findings highlight the important role of DNA repair in human tissues and cancer risk, and might inform future strategies for tumor prevention in XP-C patients.

[1] INSERM U981, Gustave Roussy Cancer Campus, Université Paris Saclay, Villejuif, France. [2] Department of Children and Adolescents Oncology, Gustave Roussy, Villejuif, France. [3] Université Paris-Saclay, UVSQ, Inserm, CESP, Villejuif, France. [4] Centre de Recherche des Cordeliers, « Equipe labélisée Ligue Contre le Cancer », CNRS SNC 5096, Sorbonne Université, Université de Paris Cité, INSERM, Paris, France. [5] Université de Paris Cité, Paris, France. [6] Institut du Cancer Paris CARPEM, AP-HP, APHP Centre, Department of Gynecological Surgery, Hôpital Cochin, Paris, France. [7] Liberal Endocrinologist, Algiers, Algeria. [8] Departments of Pathology, Gustave Roussy, Villejuif, France. [9] CNRS UMR9019 Genome Integrity and Cancers, Gustave Roussy, Université Paris-Saclay, Villejuif, France. ✉email: sergey.nikolaev@gustaveroussy.fr

Xeroderma pigmentosum (XP) is a group of rare hereditary recessive disorders characterized by UV hypersensitivity and up to 10,000 times increased risk of skin cancers due to the inability of cells to repair bulky UV-induced DNA lesions. Recently, we showed that XP patients have 34-times increased risk of internal tumors versus general population[1]. XP group C (XP-C) patients are characterized by defects of Global-Genome Nucleotide Excision Repair (GG-NER) and are particularly prone to early onset of some internal cancers, such as leukemia, cancers of central nervous system, thyroid and gynecological cancers[1]. Analysis of hematological malignancies from XP-C patients revealed specific mutation signature, 25-times increased mutation load and complex karyotype with predominant TP53-mutations[2].

Here, we report five gynecological malignancies and their detailed genomic characterization from three young female XP-C patients. We reveal mutated *DICER1* gene in all tumors and significantly increased mutagenesis with specific mutation signature characteristic of XP-C patients and GG-NER deficiency.

## Methods

**Ethics approval and consent to participate**. This study was approved by the French Agency of Biomedicine (Paris, France), the Ethics Committee from the CPP of the University Hospital of Bordeaux (Bordeaux, France), the Institutional Review Board of Gustave Roussy (CSET: 2018–2820; Gustave Roussy, Villejuif, France). Informed signed consents to publish the case details including identifiable information were obtained from patients and/or their parents according to the Declaration of Helsinki and the French law.

**Processing of the tumor and germline samples**. The treatment-naïve tumor samples were collected from patients during surgery. The tumors were stored in liquid nitrogen or allprotect tissue reagent. Normal control samples were represented by blood or fibroblast cell line. DNA was extracted using AllPrep DNA/RNA/miRNA Universal Kit (Cat. No. / ID: 80224, Qiagen) according to the manufacturer's instructions. DNA quantity and quality were assessed using the NanoDrop-ND-1000 (Nanodrop Technologies).

**Genome sequencing and bioinformatic processing**. The genomes were sequenced using BGISEQ-500 in BGI (Shenzhen) according to the manufacturer's protocols (mean coverage equal to 73X for tumor and 37X for normal DNA). Reads were mapped to the GRCh37 human reference genome using BWA-MEM[3,4] (v0.7.12) software, and then processed with GATK best practice pipeline[5] to call somatic and germline genetic variants (GATK[6] v4.0.10.1). Somatic variants were annotated with oncotator[7] (v1.9.9.0) and oncogenic mutations were inferred using OncoKB database[8] (MafAnnotator.py v.3.3.0, https://github.com/oncokb/oncokb-annotator). SCNAs calling was done with FACETS[9] (v 0.5.14). Germline genetic variants were annotated with ANNO-VAR database[10]. Ovarian granulosa cell tumors (OGCT) from sporadic patients[11] were processed from the stage of FASTQ files in the similar way as XP-C tumors.

Only somatic variants with PASS flag supported by at least one read from each strand and at least three reads in total with variant allele frequency above 0.05 were used for the analysis for all the samples.

**Mutation signature analysis**. We used SigProfilerMatrixGenerator v.1.0 software[12] to convert the VCF files into a catalog of mutation matrices. We computed pairwise Cosine similarity distance between all the samples using MutationalPatterns R package[13,14] (cos_sim_matrix()) and then processed the matrix of distances between the samples in the prcomp() function in R to obtain multidimensional scaling plots (MDS). Mutation signature deconvolution was performed with SigProfilerExtractor[15].

**Statistics and reproducibility**. We used nonparametric two-sided Mann–Whitney $U$ test to compare tumor mutational burden of single base substitutions (SBS), double base substitutions (DBS) and indels between sporadic OGCT ($n = 45$), XP-C leukemia samples ($n = 6$) and XP-C ovarian tumors ($n = 3$). All the tumors were independent biological replicates.

**Reporting summary**. Further information on research design is available in the Nature Portfolio Reporting Summary linked to this article.

## Results

**Case description: patients 1 and 2**. Monozygotic twin sisters (XP2003VI and XP2004VI) were diagnosed with XP during their first year of life due to very high sun-sensitivity. Genetic testing revealed homozygous c.1643_1644delTG; p.Val548Alafs25 (called therefore delTG) mutation in the *XPC* gene, thus confirming the diagnosis (Table 1, Fig. 1a). The patients were not diagnosed with skin cancers but had actinic keratosis and freckles on their faces. The patients were from consanguineous family. For both 16 years old-patients, following hemorrhages a large cervical mass that was prolapsing through the vagina, was discovered following pelvic MRI with no local extension. Histological examination of the biopsies confirmed a vaginal embryonal rhabdomyosarcoma (vERMS). The tumoral cells were positive for Desmin and Myogenin expression. These tumors were classified Group C following the RMS 2005 protocol.

Patient 1: The treatment was performed according to the SFCE RMS 2005 study[16], Group C with five cycles of Ifosamide-Vincristine-Actinomycin (IVA) followed by four cycles of Vincristine-Actinomycin (Standard of Care (SoC) therapy). Chemotherapy doses have been reduced between 30 and 50% because of their toxicity due to the patient DNA repair-deficiency[17,18]. Following the chemotherapy and after ovarian transposition, local treatment was performed with vaginal brachytherapy with a delivered dose of 24 Gy. The patient recovered completely and is followed for her XP disease regularly without any specific medical problems for total 10 years follow-up.

Patient 2: The same SoC treatments with chemotherapy, surgery and brachytherapy as her sister were administered to this patient at the same age (Table 1). At the age of 20, pancytopenia was discovered following standard exams for XP with 7% of blast cells in blood and 74% of bone marrow blast cells. Immunophe-notyping revealed an acute myeloid leukemia type 2 (AML-2). Treatment with Adriamycin and Aracitidine for 4 months induced a complete remission but a year later the patient developed a total aplasia and died at 22.

**Case description: patient 3**. Patient 3 (XP694VI) is a 19-years old female diagnosed with XP-C at the age of three and had a history of skin carcinomas, ephelides and freckles on the sun-exposed body sites (Table 1). She was referred for an ACTH-independent Cushing's syndrome leading to the discovery of a rapidly growing left ovarian tumor of 11 cm. A laparotomy was performed. On surgical exploration, a preoperative cyst rupture was found, without any suspicious lesion on the peritoneum. A left adnexectomy was decided. Expert pathologic examination concluded to a juvenile granulosa-cell tumor of the ovary (J-GCT), partially ruptured, with a negative peritoneal cytology but tumor cells on the tubal serosa. The patient was treated with the SoC therapy with slight modifications due to DNA repair

**Table 1 Clinical profiles of the patients.**

| Patient's code | P1 | P2 | P3 | P4 |
|---|---|---|---|---|
| ID | XP2003VI | XP2004VI | XP694VI | XP2020VI |
| XPC gene mutation | V548fs | V548fs | V548fs | V548fs/E284X |
| Age of XP diagnosis (years old) | 1 | 1 | 3 | 2 |
| Origin of patient's families | Algeria | Algeria | Algeria | Tunisia |
| Skin cancers | No | No | A few carcinomas | No |
| Other cutaneous manifestations | Actinic keratosis, freckles on the face | Actinic keratosis, freckles on the face | Ephelides, freckles on the exposed body sites | Melanocytic hyperplasia |
| Gynecological tumors (age, years) | vERMS (16) | vERMS (16) | J-GCT of the left ovary (19) and poorly differentiated stage IA SLCT on the right ovary (19) | Poor differentiated Sertoli-Leydig cell tumor with heterologous component and retiform pattern of the left ovary - FIGO stage IC1 (11) |
| Treatment | 5 cycles of Ifosamide-Vincristine-Actinomycin (IVA) followed by 4 cycles of Vincristine-Actinomycin. Brachytherapy (24 Gy) | 5 cycles of Ifosamide-Vincristine-Actinomycin (IVA) followed by 4 cycles of Vincristine-Actinomycin. Brachytherapy (24 Gy) | Adnexectomy | Left adnexectomy - No adjuvant chemotherapy contrary to recommendations due to the high risk of toxicity of cisplatinum in XP patients |
| Treatment response | Complete response | Complete response for the vERMS | SLCT: complete response; J-GCT: recurrence 9 months later treated by colorectal resection and paclitaxel carboplatin; progression after 4 cycles chemo; | Complete response |
| Total follow-up after gynecological cancer treatment | 11 years | 6 years | 9 months | 32 months |
| Other internal cancers | No | AML-2 at 20 years of age Treatment by Adriamycin and Azacitidine | No | No |
| Status in 2023 | Alive | Death at 22 years of age due to AML-2 | Alive | Alive |

vERMS vaginal embryonal rhabdomyosarcoma, AML-2 acute myeloid leukemia type 2, J-GCT juvenile granulosa-cell tumor of the ovary, SLCT Sertoli-Leydig cell tumor.

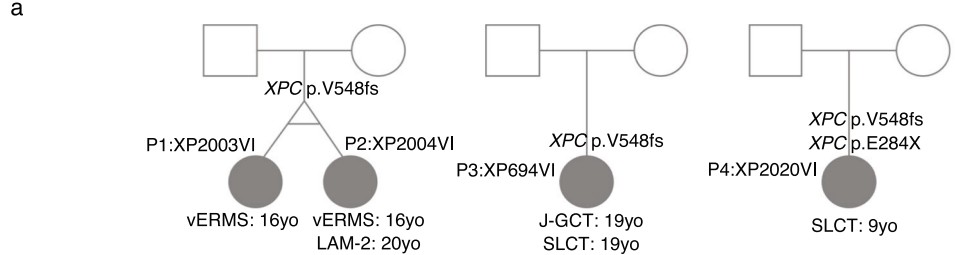

**Fig. 1 Genomic analysis of the XP-C gynecological tumors. a** Family pedigree of the studied patients. Germinal pathogenic variants and diagnoses are indicated. vERMS - vaginal embryonal rhabdomyosarcoma, LAM-2 - acute myeloid leukemia type 2, J-GCT – juvenile granulosa-cell tumor of the ovary, SLCT - Sertoli-Leydig cell tumor. **b** Tumor mutational burden of single base substitutions (SBS), double base substitutions (DBS) and indels in sporadic Ovarian Granulosa Cell Tumors ($n = 45$, blue boxplot), XP-C leukemia samples ($n = 6$, orange boxplot) and XP-C ovarian tumors ($n = 3$, red boxplot; $P$-values are indicated, Mann–Whitney $U$ test, two-sided). All the tumors were independent. Boxes depict the interquartile range (25–75% percentile), lines - the median, whiskers—1.5× the IQR below the first quartile and above the third quartile. **c** Multidimension scaling plots (MDS) for dimensions 1 and 2 and 1 and 3 based on the cosine similarity distances between the SBS trinucleotide-context mutation profiles of individual tumors. XP-C tumors and XP-C iPSCs form a separate group. OGCT—ovarian granulosa cell tumors, iPSCs—induced pluripotent stem cells. **d** The mean (SE intervals are indicated) transcriptional bias per tumor type (ratio between untranscribed and transcribed strand) for each major type of substitutions ($n = 45$ for sporadic OGCT, $n = 6$ for XP-C leukemia, $n = 3$ for XP-C ovarian independent tumors). **e** Fraction of mutations for each tumor explained by different COSMIC mutational signatures. **f** Oncogenic or likely oncogenic mutations in XP-C gynecological tumors according to oncoKB database (m – missense mutation, f – frameshift mutation, s – stop codon gain, n – nonsense mutation, a - promoter activating mutation).

deficiency. Adjuvant chemotherapy, despite being theoretically indicated for a FIGO stage IC2 J-GCT, was rejected because of an increased risk of adverse events in the context of XP-C. Restaging laparoscopy was decided and found no tumor localization on all biopsies. However, hypercortisolism was still persistent three months after surgery. A magnetic resonance imaging found a new right ovarian mass of 2.5 cm and a micronodular left adrenal gland. Right adnexectomy and left adrenalectomy were performed by laparoscopy and histological examination found a poorly differentiated stage IA Sertoli-Leydig cell tumor (SLCT) on the right ovary and a primary pigmented nodular adrenal dysplasia. Following the last operation, Cushing's syndrome has improved and there have been no recurrences during 6 months follow-up. After nine months of follow-up, a magnetic resonance imaging found a bulky pelvic tumor. A subsequent laparotomy was performed, and the mass was completely removed, requiring a colorectal resection. Pathologic examination confirmed J-GCT recurrence. She received four injections of weekly paclitaxel, but adjuvant treatment had to be modified due to anxiety disorders. She continued receiving paclitaxel 175 mg/m2 every 3 weeks for three cycles without experiencing serious adverse events.

**Case description: patient 4**. Patient 4 (XP2020VI) is a 13-years old female with XP-C diagnosed at the age of two. As usual in XP, she had a history of melanocytic hyperplasia (Table 1). At the age of 11, before puberty onset, the patient suffered from recurrent abdominal pain, abdominal enlargement and slight vaginal bleeding. On MRI, a large cystic and solid abdominal mass (larger axis equal to 20 cm) was observed suggesting an ovarian tumor. AFP was slightly increased (64.8 ug/l) whereas HCG, Inhibin B and AMH were normal. A left adnexectomy with per-operative cystic rupture was performed with peritoneal washing and peritoneal examination. Final diagnosis was ovarian FIGO stage IC1 SLCT with predominantly retiform pattern and heterologous components, inhibin positive. Peritoneal washing was cytologically negative and no distant lesion was observed. AFP level normalized after surgery. Because of the great sensitivity of XP patients to chemotherapy and especially to cisplatin[17], the main chemotherapy agent used in this kind of tumor, and contrary to actual national guidelines recommending systematic adjuvant chemotherapy with three cycles of BEP (Bleomycin-Etoposide-Cisplatin), it was decided to not give adjuvant chemotherapy to this young XP child in agreement with her parents. Other treatments were performed according to the SoC. The patient is currently still followed every 3 months with abdominal and pelvic MRI and biological markers (Inhibin B and AFP). During the 20 months-long follow-up no recurrences have been detected.

**Genomic analysis**. For patients 1–4 we obtained tumor and/or normal DNA samples and performed Whole Genome or Exome Sequencing (WGS or WES). Patient 1: WGS of tumor (vERMS - AS2005) and germline DNA[2], Patient 2: WES of germline DNA (this study), Patient 3: WGS of 2 tumors (SLCTs - SA015T1, and J-GCT - SA015T2) and germline DNA (this study), Patient 4: WGS of tumor (SLCTs - SA014T1) and germline DNA (this study).

**Germinal DNA analysis**. Analysis of germinal DNA of patients revealed abundance of runs of homozygosity (RoH) in patients 1–3 (27–65 Mbp per individual) including RoH overlapping *XPC*, and a causative founder homozygous germline mutation typical for the studied Northern African XP-C population (delTG, *XPC* p.V548fs). Patient 4 had a compound heterozygous mutation in *XPC* gene (p.E284X and p.V548fs) and only 12 Mbp in RoHs across the genome (Fig. 1a). We did not identify any

cancer-predisposing germline variants except in the *XPC* gene in the studied patients.

In twin sisters (patients 1 and 2) with young-onset vERMS we identified a region of homozygosity on chromosome 19 (7.5–13.2 Mbp) which overlaps *SMARCA4*. Mutations in this gene are known to be associated with Rhabdoid Tumor Predisposition Syndrome 2[19] (RTPS2). However, we did not identify any mutations in *SMARCA4* in our patients. Moreover, the IHC staining for SMARCA4 revealed normal level of protein expression. These results did not confirm RTPS2 in patients 1 and 2.

**Analysis of somatic mutations in tumors**. To understand genetic basis of gynecological tumors in XP-C patients we compared WGS data from those 4 cancers to other 7 internal cancers from XP-C patients[2], and to a panel ($n = 45$) of diverse OGCT[11]. Genomic mutation analysis revealed a mutator phenotype in XP-C gynecological cancers with a mean of 27216 mutations per genome. Mutational density in XP-C gynecological tumors was 4.4-fold increased ($P = 0.00046$, Mann–Whitney $U$ test, two-sided) in comparison with sporadic OGCT for SBS and 1.7-fold increase for indels ($P = 0.041$, Mann–Whitney $U$ test, two-sided), but was not significantly different from the other types of internal XP-C tumors of hematological origins (Fig. 1b).

To better understand if contribution of XPC deficiency to the mutational processes is confined to cancer or also impacts normal tissues we combined our WGS cancer dataset with WGS of iPSCs from $n = 20$ XP samples and $n = 20$ controls[20]. We compared mutational profiles between individual genomes using cosine similarity distance and revealed clear separate grouping of XP-C gynecological tumors with XP-C leukemias, XP-C breast sarcoma and with XP-C iPSCs on the MDS plot (Fig. 1c). In line with previous observation in XP-C leukemia, in XP-C gynecological tumors we revealed strong transcriptional bias across 6 main substitution groups (C > A/T/G,T > A/C/G; Fig. 1d). We further refitted known COSMIC mutational signatures to our dataset and revealed an enrichment of the SBS8 signature in XP-C ovarian tumors and rhabdomyosarcoma (67–77% of mutations explained; mean = 71%) and in 3/5 XP-C iPSCs (Fig. 1e). In sporadic cohort of OGCT tumors SBS8 explained only 0-38% (mean = 23%) of mutations and dominating mutational signature was SBS5 usually widespread across different tumor types and normal human tissues[21].

Analysis of potential oncogenic events in the four studied tumors revealed oncogenic or likely oncogenic mutations in *DICER1* gene in all cases, with biallelic deactivation in 2 tumors. *TP53* was mutated in vERMS and in SLCT tumor from patient 3. Genomes of J-GCT and SLCTs were relatively stable without polyploidization, while the genome of vERMS was polyploid and unstable with focal amplifications of *RICTOR* (16 copies) and *GLI2* (14 copies) genes (Fig. 1f).

**Discussion**

Here, we described 5 cases of gynecological tumors and for 3 XP patients with detailed genomic analysis. All of them represented XP-C group with the founder delTG mutation common across the Northern Africa[1]. This is in line with our recent study where we revealed a 135 (95%CI 53–329) times increased risk of gynecological cancers in XP-C population with delTG mutation versus the general population[1]. Using the literature search we identified 11 previously published reports of gynecological tumors in XP patients (Table 2). The median age of the patients was 18.5 years with diverse presentation of diagnoses including uterine carcinoma, cervical sarcoma, uterine leimyosarcoma, sex-cord stromal tumors and other gynecological malignancies. All the

**Table 2 Gynecological tumors reported in xeroderma pigmentosum patients.**

| Tumor types[a] (patient code) | Country of patient's origin | Age at tumor diagnosis (years) | Age at death (years) | Gene mutation | References |
|---|---|---|---|---|---|
| Uterine carcinoma | Japan | 49 | 51 | ni | Satoh Ok& Nishigori 1988[30] |
| Uterine adenocarcinoma (XP1BE) | USA | ni | 49 | XPC: c.1132_1133 delAA | Khan et al 2006[31] |
| Cervical sarcoma (XP269VI) | Morocco | 18 | 23 | XPC: delTG[b] | Hadj-Rabia et al 2013[32] |
| Uterine leiomyosarcoma | Tunisia | 19 | 19 | XPC: delTG | Jerbi et al 2016[33] |
| Uterine leiomyosarcoma | Tunisia | 22 | 29 | XPC: delTG | Jerbi et al 2016[33] |
| Ovarian mature cystic teratoma | China | 28 | ni | XPC: HZ c.2218_2220delCTC; p.Glu740_740del HZ c.2257dupC; p.Arg753fs | Tang et al 2018[34] |
| Serous ovary Carcinoma | Brazil | 27 | Alive | XPC: delTG | Santiago et al 2020[35] |
| Uterine rhabdomyosarcom[c] (XP2004VI) | Algeria | 16 | 22 | XPC: delTG | Yurchenko et al 2020[2] |
| Uterine rhabdomyosarcom[c] (XP2003VI)[d] | Algeria | 16 | Alive | XPC: delTG | This work |
| Ovarian sarcoma (XPEIHaVI) | Morocco | 18 | 22 | XPC: delTG | Nikolaev et al 2022[1] |
| Uterine adenosarcoma (XPEIKaVI)[d] | Morocco | 15 | Alive | XPC: delTG | Nikolaev et al 2022[1] |
| Sertoli-Leydig cell tumor (XP2O2OVI) | Tunisia | 11 | Alive | XPC: HZ delTG HZ c.G850T; p.E284X | This work |
| 2 independent ovarian tumors in the same patient: (a) Juvenile granulosa-cell tumor of the ovary (left) (b) Sertoli-Leydig cell tumor (right) (XP694VI) | North Africa | 19 | Alive | XPC: delTG | This work |
| Sertoli-Leydig tumor left | Tunisia | 27 | Alive | ni | Bdioui et al 2021[36] |
| Sex-cord stromal tumor | Tunisia | 7 | Dead | ni | Bacha et al 2019[37] |

*ni* not indicated.
[a]The tumor types are expressed as exactly reported in the indicated publications.
[b]delTG refers to the North-African founder mutation: XPC c.1643_1644delTG; p.Val548AlafsX572.
[c]Monozygotic twins.
[d]Sisters.

patients with reported germline variants were of XP-C group and the majority originated from the Northern Africa harboring the founder *XPC* delTG mutation. Among our patients, two cases are of particular interest: twin monozygotic sisters who simultaneously developed vERMSs at their sixteenth (patients 1 and 2) and patient 3 who developed two genetically independent ovarian tumors at 19 years of age. This coincidence additionally highlights the risks for early onset gynecological tumors in XP-C patients.

The results of WGS revealed significantly increased mutagenesis in ovarian tumors of XP-C patients and enrichment of mutational signature SBS8 which was previously associated with the nucleotide excision repair deficiency by our group[2] and others[22]. The biochemical etiology and endogenous chemical agents which induce this signature are not known, but the strong transcriptional bias suggests an involvement of bulky purine DNA lesions[2]. These lesions are efficiently repaired by the transcription-coupled NER on the transcribed strand of genes but are not repaired in the rest of the genome in the absence of XPC protein causing accumulation of mutations on the untranscribed relative to transcribed gene strands. The same mutational signature was enriched in the erythroblast-derived XP-C iPSCs from Insignia project[20], but interestingly not in other XP iPSCs. Moreover, very high rate of de-novo mutations associated with SBS8 and transcriptional bias was recently reported in a child of an XP-C male patient, confirming that XPC deficiency driven mutagenesis may affect multiple tissue types[23].

Our genomic analysis identified mutated *DICER1* in all four studied tumors. *DICER1* mutations are quite common in uterine ERMS[24] and also in SLCT[25]. Specific mutational signature of XP-C cancers together with high mutation load can favor occurrence of *DICER1* mutations leading to gynecological malignancies of XP. Larger sample sizes will be needed to better understand the role and specificity of *DICER1* mutations for XP gynecological tumors. The genomes of the ovarian tumors were relatively stable and diploid. The vERMS tumor from patient 1 demonstrated at least five oncogenic events and genome polyploidization.

vERMSs of patients 1 and 2 were treated with reduced dosage of chemotherapy which led to the complete remission of patient 1 but could trigger development of AML-2 in patient 2. As leukemias are frequent among untreated XP-C patients, without possible analysis of patient's AML-2 sample it is hard to assess the causative role of chemotherapy in this case[1,2,26]. Patients 3 (J-GCT and SLCT) and 4 (SLCT) benefitted from the treatment without adjuvant chemotherapy because of their XP-C status despite the clinical recommendations for such disease presentation. These cases show that DNA repair deficiency should be accounted for before the treatment to minimize adverse effects in mostly young XP patients.

In this work, we demonstrated increased mutagenesis, specific mutation signature and common *DICER1* alterations in gynecological tumors of XP-C patients. Our results point to the genomic basis of gynecological tumor susceptibility in XP-C patients caused by increased mutagenesis due to inability to repair probably endogenous bulky purine DNA lesions. We suggest to regularly follow young XP-C patients by gynecologists to timely identify gynecological (pre-)malignancies and adjust adjuvant chemotherapeutic treatment according to the DNA repair status of patients[17,18]. It was recently shown that skin cancers in XP patients can be very efficiently treated with immunotherapy due to the high level of mutagenesis[27]. Although the trials on immunotherapy in rare gynecological malignancies are in their infancy[28,29], this study of XP gynecological cancers, together with our previous work on XP leukemias indicate that a high level of mutagenesis (a marker of response to the immunotherapies in solid tumors; FDA Level 2) is a characteristic for XP-C internal tumors. We believe that sequencing larger collections of internal tumors from XP patients can intensify the development of new immunotherapy regiments for this vulnerable group.

## Data availability

The dataset generated during the current study (WGS FASTQ files and corresponding somatic VCF files) is available from the corresponding author on reasonable request after approval by the data access committee due to the data privacy of the patients. The source data to reproduce the figures is available in Supplementary Data 1 file. Genomic datasets of XP-C leukemia and sporadic OGCT tumors used in this study are available in the European Genome-Phenome Archive (EGA) under accession codes EGAS00001004511 and EGAS00001004249 respectively, access is restricted and can be granted under approval by the data access committees.

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

## Acknowledgements

S.I.N. was supported by grant Foundation ARC 2017, Foundation Gustave Roussy, and the French National Cancer Institute - RPT21145LLA. A.S. is particularly thankful to the French XP Association "Les Enfants de la Lune" and the parents of XP patients for their support. We also thank very much the different dermatologists, who are following the described XP patients, for their help in their clinical description.

## Author contributions

S.I.N. and A.S. planned the study. A.Y. performed genomic analysis and prepared figures. A.Y., A.S., B.F., B.B. and S.N. drafted the manuscript. F.J. handled samples and extracted DNA. B.B., Z.T. and B.F. were involved in the clinical care of the patients and provided relevant material. C.G. performed pathological examination of the materials.

## Competing interests

The authors declare no competing interests.
