## [Peer Review File · Communications Medicine]

Reviewers' comments:

Reviewer #1 (Remarks to the Author):

In this manuscript, Yurchenko et al. evaluated the genomic profile (germline and somatic) of 5 gynecological tumors from XP-C patients. They found a DICER1 mutation in all tumors and a specific mutation signature of XP-C patients and GG-NER deficiency. The clinical description was very well-detailed for all patients. This study also highlights that patients harboring DNA repair deficiency should be accounted for before the treatment to minimize adverse effects, mostly in young patients. This is a well-documented study with findings of the impact that increase our knowledge of this rare disease.

I have a few comments:

1. Reference 2: should be presented with complete information.

Yurchenko AA, Padioleau I, Matkarimov BT, Soulier J, Sarasin A, Nikolaev S. XPC deficiency increases risk of hematologic malignancies through mutator phenotype and characteristic mutational signature. *Nat Commun.* 2020 Nov 17;11(1):5834. doi: 10.1038/s41467-020-19633-9. PMID: 33203900; PMCID: PMC7672101.

2. It is curious that DICER1 mutation was found in all tumors of the 4 XP-C patients. Although the authors stated that this event is common in ERMS and SLCT, having a hypothesis or plausible explanation for these findings will be interesting.

Reviewer #2 (Remarks to the Author):

This manuscript is a case report on 5 early-onset gynecological tumors in 4 xeroderma pigmentosum (XP) patients. One of the greatest strengths of this paper is the report on monozygotic twin sisters who carried homozygous c.1643_1644delTG; p.Val548Alafs25 pathogenic variant in the XPC gene and who developed embryonal rhabdomyosarcoma (vERMS) both at age 16 is of great interest. Both had similar clinical presentation and shared the same histological analysis. Nevertheless, recovery was complete to one of the twins, with a follow-up of 10 years, whereas the other twin developed an acute myeloid leukemia at the age of 20, a year later total aplasia and died at 22. This is a novel description of identical twins carrying the same germline XPC pathogenic variant and it represents a perfect human model that might provide insights on the understanding on what are the underlying interacting genetic factor that lead XPC carriers of the same mutation develop the exact same tumor at the same age.

A minor limitation need to be addressed. The manuscript has a poor clinical presentation of all included cases. A thorough description on tumor spectrum and age of onset should be provided as well as the description of other typical features of the syndrome.

The work is convincing and it is a very well written paper suitable for publication after the addition of a table containing the clinical profile of all 5 cases.

Reviewers' comments:

Reviewer #1 (Remarks to the Author):

In this manuscript, Yurchenko et al. evaluated the genomic profile (germline and somatic) of 5 gynecological tumors from XP-C patients. They found a *DICER1* mutation in all tumors and a specific mutation signature of XP-C patients and GG-NER deficiency. The clinical description was very well-detailed for all patients. This study also highlights that patients harboring DNA repair deficiency should be accounted for before the treatment to minimize adverse effects, mostly in young patients. This is a well-documented study with findings of the impact that increase our knowledge of this rare disease.

I have a few comments:

1. Reference 2: should be presented with complete information.

Yurchenko AA, Padioleau I, Matkarimov BT, Soulier J, Sarasin A, Nikolaev S. XPC deficiency increases risk of hematologic malignancies through mutator phenotype and characteristic mutational signature. *Nat Commun.* 2020 Nov 17;11(1):5834. doi: 10.1038/s41467-020-19633-9. PMID: 33203900; PMCID: PMC7672101.

Response: corrected.

2. It is curious that *DICER1* mutation was found in all tumors of the 4 XP-C patients. Although the authors stated that this event is common in ERMS and SLCT, having a hypothesis or plausible explanation for these findings will be interesting.

Response: we thank the reviewer for this comment and agree that presence of *DICER1* mutations in all the tumor samples is an interesting result. Unfortunately, the low number of cases with genomic analysis preclude us from making a strong hypothesis. There is a possibility that a specific mutational process observed in the cancers of XPC patients can favor *DICER1* damaging mutations which are important in gynecological

cancers but we do not have a strong evidence for that yet. Encouraged by the reviewer we shortly expanded the Discussion section (L262-266).

Reviewer #2 (Remarks to the Author):

This manuscript is a case report on 5 early-onset gynecological tumors in 4 xeroderma pigmentosum (XP) patients. One of the greatest strengths of this paper is the report on monozygotic twin sisters who carried homozygous c.1643_1644delTG; p.Val548Alafs25 pathogenic variant in the XPC gene and who developed embryonal rhabdomyosarcoma (vERMS) both at age 16 is of great interest. Both had similar clinical presentation and shared the same histological analysis. Nevertheless, recovery was complete to one of the twins, with a follow-up of 10 years, whereas the other twin developed an acute myeloid leukemia at the age of 20, a year later total aplasia and died at 22. This is a novel description of identical twins carrying the same germline XPC pathogenic variant and it represents a perfect human model that might provide insights on the understanding on what are the underlying interacting genetic factor that lead XPC carriers of the same mutation develop the exact same tumor at the same age. \

Response: we thank the reviewer for the encouraging summary of our work.

A minor limitation need to be addressed. The manuscript has a poor clinical presentation of all included cases. A thorough description on tumor spectrum and age of onset should be provided as well as the description of other typical features of the syndrome.

The work is convincing and it is a very well written paper suitable for publication after the addition of a table containing the clinical profile of all 5 cases.

Response: we thank the reviewer for this important point and suggestion. Now we added a Table 1 (L490) which summarizes the clinical profiles of the patients as well as additional information to the text of the manuscript (L109-110, L134-135, L160).

REVIEWERS' COMMENTS:

Reviewer #1 (Remarks to the Author):

I have no further comments. This manuscript presented interesting and informative results of XP patients. I recommend the publication of this study.

Reviewer #2 (Remarks to the Author):

Authors have adequately responded to all questionings and have added Table 1 which summarizes the clinical profiles of the patients as well as additional information to the text of the manuscript. The manuscript reads very well, results are of importance in a field where information on XP is scarce and it addresses a population which is extremely rare.